

# Contribution of intermediate volatility organic compounds from on-road transport to secondary organic aerosol levels in Europe

Stella E. I. Manavi[1,2] and Spyros N. Pandis[1,2]

[1] Department of Chemical Engineering, University of Patras, Patras, GR 26540, Greece

[2] Institute of Chemical Engineering Sciences, Foundation for Research and Technology-Hellas, Patras, GR 26540, Greece

*Correspondence to*: Spyros N. Pandis (spyros@chemeng.upatras.gr)

**Abstract.** Atmospheric organic compounds with an effective saturation concentration ($C^*$) at 298 K between $10^3$ and $10^6$ μg m$^{-3}$ are called intermediate volatility organic compounds (IVOCs) and they have been identified as important secondary organic aerosol (SOA) precursors. In this work, we simulate IVOCs emitted from on-road diesel and gasoline vehicles over Europe with a chemical transport model (CTM), utilizing a new approach, where IVOCs are treated as lumped species that preserve their chemical characteristics. This approach allows us to assess both the overall contribution of IVOCs to SOA formation and the role of specific compounds. For the simulated early summer period, the highest concentrations of SOA formed from the oxidation of on-road IVOCs (SOA-iv) are predicted for major European cities, like Paris, Athens, and Madrid. In these urban environments, on-road SOA-iv can account for up to a quarter of the predicted total SOA. Over Europe, unspeciated cyclic alkanes in the IVOC range are estimated to account for up to 72% of the total on-road SOA-iv mass, with compounds with 15 to 20 number of carbons being the most prominent precursors. The sensitivity of the predicted SOA-iv concentrations to the selected parameters of the new lumping scheme is also investigated. Active multigenerational aging of the secondary aerosol products has the most significant effect as it increases the predicted SOA-iv concentrations by 67%.

## 1. Introduction

Atmospheric particles, especially fine particles with a diameter of 2.5 μm or less (PM$_{2.5}$), have significant adverse effects on human health (Chen and Hoek, 2020; Pond et al., 2021) and also affect the energy balance of our planet. Organic aerosol (OA) is often the most important constituent of PM$_{2.5}$ (Zhang et al., 2007). When OA is directly emitted in the particulate phase it is called primary (POA), whereas when it is formed through the condensation of oxidation products of organic vapors it is characterized as secondary (SOA). Volatile organic compounds (VOCs) have been traditionally considered as the main SOA precursors, and their role has been the topic of studies for more than 50 years. Although atmospheric measurements have shown that SOA often comprises 60-90% of OA, air quality models often underestimate both the concentrations and the oxidation state of SOA (Zhang et al., 2007; Zhang and Seinfeld, 2013; Couvidat et al., 2013; Hayes et al., 2015, An et al., 2023). Several studies have proposed that accounting for the formation of secondary PM from the



oxidation of intermediated volatility compounds (IVOCs) can bridge the gap between measured and model-predicted SOA concentrations (Robinson et al., 2007; Pye and Seinfeld, 2010; Barsanti et al., 2013; Ait-Helal et al., 2014; Jathar et al., 2014; Ots et al., 2016; Zhao et al., 2016; Giani et al., 2019; Lu et al., 2020; Ling et al., 2022, An et al., 2023).

IVOCs are atmospheric organic gases with an effective saturation concentration ($C^*$) at 298 K between $10^3$ and $10^6$ μg m$^{-3}$ and they occupy four volatility bins in the logarithmically spaced volatility basis set (VBS) (Donahue et al, 2006; Robinson et al., 2007). This volatility range roughly corresponds to compounds with 12 to 22 carbons; including intermediate length alkanes (linear, branched, cyclic), small polycyclic aromatic hydrocarbons (PAHs), aromatics and oxidized species such as phenols, furans, ketones, and esters. Smog chamber experiments have suggested that IVOCs can be an important class of SOA precursors as their yields are higher compared to those of VOCs (Chan et al., 2009; Presto et al.,
2009; Tkacik et al., 2012; Docherty et al., 2021). Moreover, IVOCs have been detected in the emissions of mobile sources, such as on-road and off-road diesel and gasoline vehicles (Schauer et al., 1999; Gordon et al., 2014; Zhao et al., 2014; 2015; 2016; Drozd et al., 2016; Tang et al., 2021), aircraft and ship engines (Huang et al., 2018; Lou et al., 2019; Su et al, 2020) and other sources, such as biomass burning (Schauer et al., 2001; Ciarelli et al., 2017b; Hatch et al., 2017; Qian et al., 2021) and consumer products (Li et al., 2018; Seltzer et al, 2021).

Until recently, a combination of measurement and modelling challenges has prevented the simulation of the SOA formed from the oxidation of IVOCs (SOA-iv) in chemical transport models (CTMs). Most existing emissions inventories do not include IVOCs as identifying and quantifying their emissions remains challenging. There are hundreds of organic compounds and their isomers in the IVOC range, which are difficult to speciate using traditional gas-chromatography techniques. The majority of the emitted IVOCs is reported as an unresolved complex mixture (UCM) of co-eluting
compounds (Schauer et al., 1999; 2001). For example, speciated compounds (n-alkanes, b-alkanes, n-alkylcyclohexanes, n-alkylbenzenes and PAHs) constituted less than 10% of the total IVOC mass emitted from on-road diesel vehicles (Zhao et al., 2015). Besides the lack of adequate emission rates, insufficient chemical characterization of the UCM limits our ability to determine necessary SOA formation parameters, like SOA yields, vaporization enthalpies, reaction rate constants, etc.

Zhao et al. (2014) developed a method to estimate the emission factors (EFs) and the chemical characteristics of the
UCM components. At first the UCM mass is separated in eleven bins (B12-B22) across the volatility range, based on the retention times of eleven n-alkanes (C12-C22), and then it is further separated into two chemical groups, one representing unspeciated branched alkanes and one representing unspeciated cyclic compounds. The same approach was used by other studies to determine the EFs of 79 speciated and unspeciated IVOCs from on-road and off-road diesel and gasoline vehicles (Zhao et al., 2015; 2016; Tang et al., 2021; Fang et al., 2021), non-road construction machinery (Qi et al., 2019), residential
solid fuel combustion (Qian et al., 2021), vessel engines (Huang et al., 2018; Lou et al., 2019; Su et al., 2020) and in field measurements (Li et al., 2019). Recently, Huo et al. (2021) analysed samples from household biomass and coal burning using two-dimensional gas chromatography-time-of-flight mass spectrometry techniques and developed a new method for further speciating the emitted UCM mass. This approach reduced the unspeciated mass to 2% of the total measured IVOCs and provided additional insights about the chemical characteristics of the UCM from these sources.





Previous modelling studies have estimated IVOC emissions based on known emissions of other species emitted from the same source. Robinson et al. (2007) estimated IVOC emissions in the eastern US by multiplying the assumed non-volatile and non-reactive POA within the emission inventory with a factor of 1.5. This 1.5 IVOC-to-POA emissions ratio is a zeroth order assumption based on measurements of diesel emissions (Schauer et al., 1999). A number of subsequent studies have used the same or different ratios to estimate IVOC emissions both from on-road transportation and from other sources

(Murphy and Pandis, 2009; 2010; Tsimpidi et al., 2010; Koo et al., 2014; Jathar et al., 2011; Ots et al., 2016; Zhao et al., 2016; Li et al., 2020; Huang et al., 2021; Wu et al., 2021). Another approach for calculating the IVOC emissions is using an IVOC-to-VOC emissions ratio (Ots et al., 2016; Sartelet et al., 2018; Hodzic et al., 2016; Jathar et al., 2014; Lu et al., 2020; Miao et al., 2021). It is also possible to use a combination of the two ratios to estimate the emissions from different sources (Hodzic et al., 2016; Jathar et al., 2017; Giani et al., 2019; Lannuque et al., 2020). The choice of IVOC-to-POA or IVOC-to-

VOC ratio for the estimation of the IVOC emissions may depend on the source of interest. For example, Zhao et al. (2015) found that in diesel vehicle emissions there is a strong correlation between IVOCs and VOCs, whereas Huang et al. (2018) suggested that in ship engines the relation between IVOCs and POA is stronger.

The fist implementations of SOA-iv formation in CTMs were based on the one-dimensional volatility basis set (1D-VBS), in which IVOCs are simulated using 4 surrogate species occupying the four highest volatility bins ($C^*$ equal to $10^3$,

$10^4$, $10^5$ and $10^6$ µg m$^{-3}$) of the VBS framework (Donahue et al., 2006; Robinson et al., 2007; Murphy and Pandis, 2009; Tsimpidi et al., 2010; Jathar et al., 2011; Fountoukis et al., 2011; Li et al., 2020). In the original 1D-VBS, each of the four IVOC surrogate species reacts in the gas phase with the OH radical forming less volatile products which can then partition to the particulate phase forming SOA-iv. Koo et al. (2014) proposed the 1.5D-VBS, a modified approach in which secondary products retain information both about their volatility and their oxidation state (O:C) (Ots et al., 2016; Ciarelli et al., 2017;

Jathar et al., 2017; Giani et al., 2019; Huang et al., 2021). In the 1.5D-VBS of Koo et al. (2014), all IVOCs are represented using a single surrogate compound, that contributes to the formation of four semi-volatile products (the same four products are also used to represent the oxidation products of the VOC precursors). By assuming that all IVOCs are only characterized by their volatility and that they follow the same oxidation path, these highly parametrized schemes cannot reproduce the chemical diversity of the individual IVOCs encountered in the atmosphere. A few studies improved the depiction of IVOC

chemistry by using surrogate species such as naphthalene (Pye and Seinfeld, 2010; Miao et al., 2021) or n-pentadecane (Ots et al., 2016) to represent all IVOCs. However, the wide range of IVOCs present in the atmosphere was still oversimplified in these studies. Lu et al. (2020) improved the simulation of IVOCs emitted from mobile sources (diesel, gasoline, and aircraft) in the 1.5D-VBS approach by substituting the single IVOC surrogate with six new lumped species, which were characterized by their volatility and their chemical characteristics. The new lumping scheme and the developed semi-empirical SOA-iv

formation parametrization were based on the experiments of Zhao et al. (2015; 2016). Including SOA-iv formed from mobile sources reduced the gap between measured and model-predicted SOA concentration and the remaining difference was attributed to IVOCs emitted from other non-mobile sources. Pennington et al. (2021) extended the lumping scheme of Lu et



al. (2020) to include IVOCs from consumer and industrial products based on the emissions inventory developed by Seltzer et al. (2021).

A number of studies have simulated the effect of IVOCs on SOA formation both over the whole European domain (Fountoukis et al., 2011; Lannuque et al., 2020) and over specific European cities and regions (Ots et al., 2016; Sartelet et al., 2018; Giani et al., 2019). Although, the incorporation of IVOCs could not completely bridge the gap between measured and predicted OA concentrations, it significantly improved the model's performance for the SOA concentrations over Europe. Lannuque et al. (2020) showed that SOA-iv produced from on-road transportation can be significant at a regional

level. In megacities like London, Ots et al. (2016) found that that diesel related IVOCs could explain on average 30% of the produced SOA on an annual basis.

      Recently, to simulate IVOCs in CTMs Manavi and Pandis (2022) proposed a new approach that considers both the SOA formation potential and the complex chemistry of atmospheric IVOCs. In their approach, IVOCs are treated as lumped species and they retain their chemical characteristics (alkanes, aromatics, polyaromatics, etc.). In their work, Manavi and

Pandis (2022) have also estimated the on-road transport emissions of the new lumped IVOCs in Europe, alongside other parameters that are necessary for the simulation of the SOA formation, such as the SOA yields and the effective vaporization enthalpy. The estimated on-road diesel and gasoline vehicle emissions utilized the experimentally measured EF's of Zhao et al. (2015; 2016) and were 8 times higher compared to emissions from the same source that were estimated using the 1.5 IVOC-to-POA factor. The new lumped alkane species had the highest diesel emissions, whereas the new lumped PAH

species had the highest gasoline emissions. Moreover, the estimated SOA yields of the new lumped species were higher compared to those used for the simulation of SOA formation from the VOCs.

      Most models simulate SOA-iv formation over Europe by utilizing a variation of the VBS framework (Fountoukis et al. 2011, Ots et al., 2016; Giani et al., 2019; Lannuque et al. (2020)). Although, the VBS framework is useful in estimating the bulk contribution of IVOCs to SOA levels, it does not provide any information about the chemical characteristics of

these compounds. Moreover, little is known about the role of IVOCs in the gas-phase chemistry of the European domain. The aim of this work is to estimate the overall contribution of IVOCs to SOA concentrations and to gas-phase chemistry over Europe together with the chemical characteristics of those species. Specifically, we present and evaluate an implementation of the lumping scheme of Manavi and Pandis (2022) in PMCAMx, a three-dimensional CTM. The new version of the model is called PMCAMx-iv and it is applied over Europe for the IVOCs emitted from on-road diesel and

gasoline vehicles. The implementation and evaluation of the new lumped species approach to other IVOC sources will be the subject of subsequent publications. The sensitivity of the SOA predictions on the details of the parametrization of the proposed lumping scheme is also investigated.





## 2. Description of chemical transport models

### 2.1 PMCAMx

PMCAMx describes the processes of horizontal and vertical advection, horizontal and vertical dispersion, gas and aqueous-phase chemistry, aerosol dynamics and thermodynamics, wet and dry deposition (Murphy and Pandis, 2009; Tsimpidi et al., 2010; Fountoukis et al., 2011). The gas-phase chemical mechanism utilized in the simulations is an extension of the SAPRC mechanism (Carter, 2010; Environ., 2013). In SAPRC, VOCs are lumped into species based on their chemical characteristics and their reaction rate constant with the hydroxyl radical ($k_{OH}$). The version of SAPRC used here, includes 237 reactions of

91 gases and 18 radicals. Alkanes in the VOC range are represented by five lumped species (ALK1, ALK2, ALK3, ALK4, ALK5), olefins by two (OLE1, OLE2), aromatics by another two (ARO1, ARO2), one species represents monoterpenes (TERP) and there is also one sesquiterpenes species (SESQ).

In this application for the aerosol processes, bulk equilibrium is assumed and the 1D-VBS approach (Donahue et al., 2006) is utilized treating both primary and secondary OA species as chemically reactive. In the base version of

PMCAMx, IVOCs are represented by the $10^3$, $10^4$, $10^5$ and $10^6$ μg m$^{-3}$ volatility bins. The four surrogate species are emitted in the gas-phase and their oxidation products can partition to the aerosol phase, forming SOA-iv. With each reaction the volatility of the oxidized vapor product is reduced by one order of magnitude and its mass is increased by 7.5% to account for the added oxygen. The aging OH reactions have a reaction rate constant of $4 \times 10^{-11}$ cm$^3$ molecule$^{-1}$ s$^{-1}$. The emissions of the 1D-VBS IVOCs are calculated using the 1.5 IVOC-to-POA emissions ratio. More specifically, the emissions of the

$C^*$=$10^3$ μg m$^{-3}$ bin are assumed to be 0.15 POA, of the $C^*$=$10^4$ μg m$^{-3}$ bin 0.4 POA, $C^*$=$10^5$ μg m$^{-3}$ bin 0.5 POA and finally $C^*$=$10^6$ μg m$^{-3}$ bin equal to 0.8 POA. More detailed information about this base version of the model can be found in Murphy and Pandis (2009), Tsimpidi et al. (2010), and Fountoukis et al. (2011).

### 2.2 The PMCAMx-iv model

PMCAMx-iv uses the IVOC lumped species approach of Manavi and Pandis (2022). The new lumping scheme is an effort to

better simulate the complex chemistry of IVOCs and the SOA-iv production in CTMs. The proposed approach, and subsequently this work, focuses only on IVOCs emitted from on-road transport and more specifically on IVOCs emitted by diesel and gasoline vehicles following the studies of Zhao et al. (2015; 2016). IVOCs from other sources are simulated using the 1D-VBS approach.

### 2.2.1 The new IVOC lumping scheme

In PMCAMx-iv, seven new lumped species that represent IVOCs are added to SAPRC. The individual speciated and unspeciated compounds reported in the studies of Zhao et al. (2015; 2016) are lumped into seven species based on their chemical type and their reaction rate constant with the hydroxyl radical ($k_{OH}$). The lumping criteria used are consistent with the philosophy and criteria of the rest of the SAPRC mechanism. The seven new lumped IVOC species include four species



(ALK6, ALK7, ALK8, ALK9) representing $C_{12}$-$C_{22}$ alkanes, one species (ARO3) representing aromatics with carbon
numbers from 11 to 22 and two species (PAH1, PAH2) representing $C_{10}$-$C_{17}$ PAHs. Details about the new lumped species
and their components can be found in Manavi and Pandis (2022). The original version of SAPRC does not explicitly account
for IVOCs; but some individual species are lumped in the simulated VOCs. Thus, to avoid double counting in the new
scheme, ALK5, which previously included n-dodecane ($C_{12}H_{26}$), is revised to include compounds with a $k_{OH}$ between 0.6
and $1.3 \times 10^{-11}$ $cm^3$ molecule$^{-1}$ s$^{-1}$ (this corresponds to undecane and its isomers).

### 2.2.2 Parametrization of the gas-phase chemistry and the SOA production

All seven lumped IVOC species that are added to the model, react individually with the hydroxyl radical (OH)
producing both volatile products, that remain in the gas phase, and less volatile products that are able to partition to the
aerosol phase forming SOA-iv. The SAPRC framework is used to simulate the volatile products, while the semi-volatile and
low volatility products are simulated following the SOA VBS scheme (Lane et al., 2008). The reaction of each of the seven
added lumped species is assumed to lead to the formation of a set of five semi-volatile products with $C^*$ of 0.1, 1, 10, 100
and $10^3$ $\mu g$ $m^{-3}$ at 298 K, to simulate SOA formation. For the simulation of the more volatile products of the oxidation
reaction of the new lumped species, it is assumed that they are the same as the ones produced from the corresponding
reactions of larger VOCs with similar chemical characteristics that are already present in the SAPRC mechanism. Further
details about the added reactions are included in Manavi and Pandis (2022).

$NO_x$-depended SOA mass-based yields ($a_i$) are used in PMCAMx-iv for the seven IVOC lumped species following
Manavi and Pandis (2022). Due to lack of appropriate experimental data, the mass-based yields of the new alkane species are
assumed to be the same under high and low $NO_x$ conditions and the mass-based yields of ARO3 are assumed to be 20%
higher of those of ARO2. Both assumptions are evaluated in the sensitivity analysis section. Moreover, in the implemented
IVOC scheme, once the five condensable products are formed, they do not undergo any further oxidation. However, for
consistency reasons PMCAMx-iv includes the multigenerational aging reactions of the five condensable products but the $k_{OH}$
for all of them is set to zero. In the sensitivity section, we also assess the effect of multigenerational aging on SOA-iv
production.

### 2.3 Model application

In this study we simulate the month of May 2008, which corresponds to the EUCAARI intensive measurement campaign
across Europe. PMCAMx, which used the 1D-VBS for all sources, has been evaluated for the studied period by Fountoukis
et al. (2011; 2014). The modelling domain and the input parameters are the same as the those in Fountoukis et al. (2011;
2014). Briefly, the modelling domain corresponds to a region of 5400 $\times$ 5832 $km^2$ over Europe, with a 36 $\times$ 36 km grid
resolution and 14 vertical layers reaching up to approximately 6 km in height. A rotated polar stereographic map projection
is used. Meteorological inputs to the model, such as horizontal wind components, vertical diffusivity, temperature, pressure,
water vapor, clouds and rainfall are generated by the WRF (Weather Research and Forecasting) model (Skamarock et al.,



2008). For the anthropogenic emissions, the GEMS emissions inventory (Visschedijk et al., 2007) is used for the gas-phase emissions and the EUCAARI Pan-European Carbonaceous Aerosol Inventory (Kulmala et al., 2009) for the particulate emissions of organic and elemental carbon. The anthropogenic emissions have been developed by TNO (Netherlands Organization for Applied Scientific Research) and they include emissions from industrial, domestic, agricultural and traffic sources. The biogenic emissions are produced by MEGAN (Model of Emissions of Gases and Aerosols from Nature) and a marine aerosol emission model (O'Dowd et al., 2008) was used for sea salt emissions.

During the EUCAARI measurement campaign, observational data were consistently collected from four ground stations located in remote and rural areas and these measurements are representative of background atmospheric conditions in Europe. The four measuring sites are located in Cabauw (Netherlands), Finokalia (Greece), Mace Head (Ireland) and Melpitz (Germany). The PMCAMx-iv model results are evaluated against the hourly measurements taken from these four sites.

### 2.3.1 IVOC emissions

In PMCAMx-iv, the emissions of the new lumped IVOC species from on-road diesel and gasoline vehicles are estimated using source specific EFs of the individual compounds included in each of the seven lumped IVOCs. The EFs of the individual compounds and the total VOCs are based on two US studies (Zhao et al., 2015; 2016). The application of these emission factors to European vehicles is a necessary assumption at this stage as there is little information about IVOCs emitted from on-road gasoline and diesel vehicles in Europe. The effect of the aforementioned assumption is estimated in the sensitivity section. To avoid double counting, the 1D-VBS IVOC emissions from on-road diesel and gasoline vehicles are removed from the inventory. We have also removed 2% of the ALK5 emissions as they correspond to n-dodecane isomer emissions, which are now included in ALK6. For the simulation of the IVOCs emitted from sources other than on-road diesel and gasoline vehicles, we utilize the standard IVOC emissions of the VBS, thus their emissions are estimated using the existing IVOC-to-POA ratio.

### 2.4 Simulation cases

Eight simulations were performed to systemically explore the performance of the new lumped species approach for the simulation of IVOCs emitted from diesel and gasoline vehicles (Table 1). The first was the "VBS scheme" simulation and it was conducted with the base version of PMCAMx using the 1D-VBS approach to simulate IVOCs emitted from the different anthropogenic sources (on-road and non-road transport, agriculture, domestic and industrial sources). For the second simulation ("base case"), we used the PMCAMx-iv model that treats IVOCs emitted from on-road diesel and gasoline vehicles with the new lumped species approach. This "base case" simulation includes the updated emission inventories over Europe for that source, the new gas-phase chemistry, and the new SOA-iv parametrization. The other six simulations are part of the sensitivity analysis of PMCAMx-iv.





**Table 1: Simulation Cases**

| Name | Model Details |
|---|---|
| VBS-scheme | PMCAMx model documented in the works of Murphy and Pandis (2009), Tsimpidi et al. (2010) and Fountoukis et al. (2011) |
| Base case | PMCAMx-iv model using the IVOC lumped species approach of Manavi and Pandis (2022) |
| IVOC emissions × 2 | Double the estimated lumped species IVOC emissions for on-road diesel and gasoline vehicles |
| No gas-phase chemistry | Oxidation reactions of the seven lumped IVOC species produce only SOA-iv |
| Multigenerational aging | Condensable products react with the OH radical with a reaction rate constant equal to $4 \times 10^{-11}$ $cm^3$ $molecule^{-1}$ $s^{-1}$ |
| MW effect | Molecular weight of the condensable products equals to 250 g $mol^{-1}$ |
| ΔH effect | ΔH of the condensable products equals to 100 kJ $mol^{-1}$ |
| Different yields | The low-$NO_x$ yields for the ALK6-ALK9 species and both the low- and high-$NO_x$ yields for ARO3 are doubled |

## 3. Results and discussion

**3.1 Concentrations and diurnal profiles of on-road transport IVOCs**

Figure 1 shows the average ground-level concentrations of the lumped IVOC species predicted by PMCAMx-iv in the "base case" test. All seven species have similar spatial distributions, with the highest concentrations located in major European cities, such as Paris, London, Madrid, and Athens and regionally in countries like Italy, the Netherlands, Poland and the UK. The highest average concentrations are predicted for ALK6 and range up to 1.1 ppb. This is expected as ALK6 is also the

230 species with the highest emissions with diesel vehicles being its major source (Manavi and Pandis, 2022). PMCAMx-iv predicts that the second highest average concentrations are those of ALK7, with values ranging up to 0.7 ppb. ALK8 and PAH1 are predicted to have similar average ground-level concentrations up to 0.4 ppb. The average concentrations of the remaining 3 species are in descending order ALK9 (0-0.1 ppb), PAH2 (0-0.04 ppb) and ARO3 (0-0.01 ppb). These three surrogate species are estimated to also have the lowest emissions from on-road diesel and gasoline vehicles.



**Figure 1: Predicted average ground-level gas-phase concentrations (in ppb) of the lumped IVOC species ((a) ALK6, (b) ALK7, (c) ALK8, (d) ALK9, (e) ARO3, (f) PAH1, (g) PAH2) by PMCAMx-iv emitted by on-road transport. Different scales are used.**





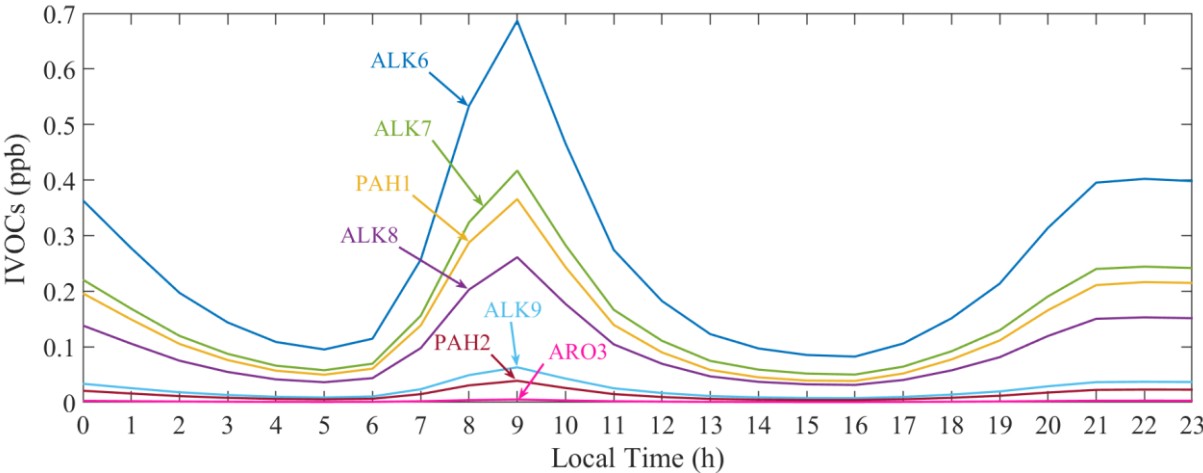

**Figure 2: PMCAMx-iv predicted average diurnal profiles for the ground-level concentrations of the lumped IVOC species emitted by on-road transport over the city of Paris for May 2008.**

The predicted diurnal profiles of the new lumped IVOC species over the city of Paris (France) for May 2008 are shown in Figure 2. PMCAMx-iv predicts that the concentrations of the seven added IVOC species start increasing after 5:00 local time (LT), leading up to the first peak of the day around 9:00 LT. This "morning" peak is also the highest predicted during the day. After the first peak, ground-level IVOC concentrations decrease and stabilize until 16:00 LT, when IVOC concentrations begin to increase again leading to the second peak of the day around 21:00 LT. The concentrations predicted during the second peak are approximately half of those predicted for the morning peak. Later in the night, IVOC concentrations decrease reaching their lowest values at 5:00 LT. The increase and the two peaks in the modelled concentrations correspond to on-road traffic in Paris. After the first peak, the predicted reduction of the lumped IVOC concentrations is both due to their oxidation reactions and to dilution as the mixing height increases. After sunset (occurring on average after 21:00 LT), the simulated chemistry slows down, the mixing height decreases and the corresponding emissions, even if they are much lower than in the morning, result in the second predicted peak.

Figure 3 compares the hourly-averaged ground-level IVOC concentrations in Paris that are predicted by PMCAMx and PMCAMx-iv. The comparison focuses only on the IVOCs emitted from on-road diesel and gasoline vehicles. The higher emissions added in the lumped IVOC scheme result in higher IVOC concentrations compared to those predicted by the "VBS scheme" simulation. On average there is one order of magnitude difference between the concentrations predicted by the two schemes. Besides the difference in the predicted values, the two timeseries are similar with the highest and lowest values of the two models coinciding. For example, for Paris both models estimate that the highest total IVOC ground-level concentration from diesel and gasoline vehicles occurs during May 16, 2008, at 10:00 LT. PMCAMx-iv predicts a concentration of 5 ppb, whereas PMCAMx a concentration of 0.5 ppb. The similarity between the two timeseries is also confirmed by the average diurnal profiles of the ground-level IVOC concentrations in Paris (Figure S1).




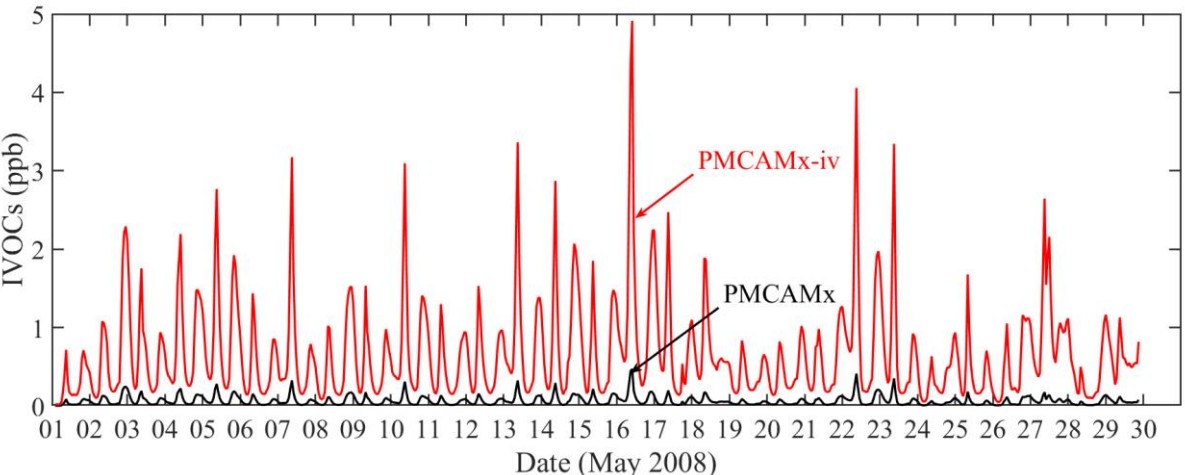

**Figure 3: Hourly-averaged ground-level IVOC concentration emitted by on-road vehicles in Paris as predicted by PMCAMx and PMCAMx-iv for May 2008.**

## 3.2 SOA-iv formation from on-road transport

### 3.2.1 Predicted SOA-iv concentrations in an urban environment

In PMCAMx-iv, SOA-iv is formed from the condensation of the five low-volatility and semi-volatile products of the seven lumped IVOC species. Figure 4 depicts the hourly-averaged ground-level concentrations of $PM_{2.5}$ SOA-iv from transportation over the city of Paris for the simulated month. For Paris, the mean on-road SOA-iv concentration that is predicted by PMCAMx-iv is 0.13 µg m$^{-3}$, while there are several days during the month that hourly-averaged concentrations reach up to 0.6 µg m$^{-3}$. PMCAMx using the 1D-VBS framework predicts lower on-road transport SOA-iv concentrations. Moreover, the two temporal profiles of the SOA-iv concentrations are different, with PMCAMx predicting the highest on-road transport SOA-iv concentration of the month at a different date (0.14 µg m$^{-3}$ on May 10) compared to the date predicted by PMCAMx-iv.

The diurnal profiles also highlight the fact that in an urban environment the on-road SOA-iv concentrations predicted using the lumped IVOCs scheme have a different temporal pattern compared to those predicted using the 1D-VBS scheme (Figure 5). PMCAMx-iv on-road transport SOA-iv concentrations start increasing around 6:00 LT, following the concentration increase of the lumped IVOC species with one hour difference, and they peak at around 9:00 LT. The peak concentration (0.16 µg m$^{-3}$) is followed by a decrease until 18:00, when concentrations begin to increase again until they reach a value of 0.13 µg m$^{-3}$ at 21:00 LT. After sunset, predicted average on-road SOA-iv concentrations are relatively stable until the early morning hours (3:00 LT) when concentrations begin to decrease, due to dilution and transport out of the city. Comparison between the on-road SOA-iv diurnal profiles of the two models shows that the lumped species approach forms SOA-iv faster than the 1D-VBS approach. In PMCAMx average SOA-iv concentrations begin increasing only after 10:00 LT, five hours after IVOC concentrations begin to increase and three hours after PMCAMx-iv SOA-iv concentrations





increase. Due to its late response, the highest concentration of SOA-iv with the 1D-VBS approach is reached in late afternoon (16:00 LT) and it is maintained until 21:00 LT. Tsimpidi et al. (2011), while evaluating the performance of PMCAMx in an urban environment (Mexico City), found that the model was too slow to reproduce accurately the high SOA concentrations that were observed during morning hours. Although, a one-to-one comparison between this study and the study of Tsimpidi et al. (2011) is not possible, the faster response of PMCAMx-iv is an encouraging result.

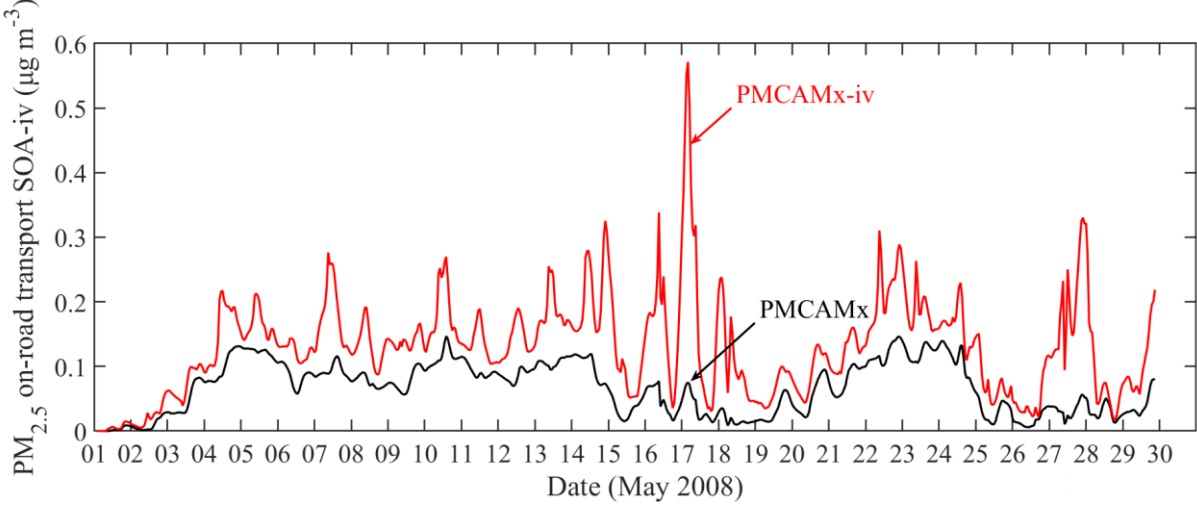


**Figure 4: PM₂.₅ hourly-averaged timeseries for the predicted ground-level concentrations of on-road transport SOA-iv by PMCAMx and PMCAMx-iv over the city of Paris for May 2008.**

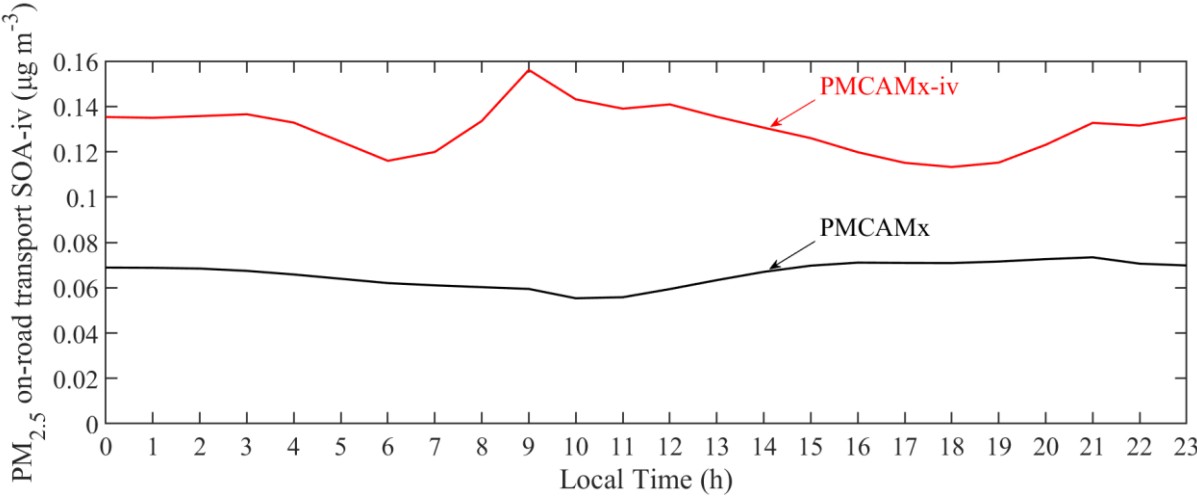

**Figure 5: Average diurnal profile of the PM₂.₅ ground-level on-road transport SOA-iv concentrations over Paris predicted by PMCAMx and PMCAMx-iv.**



### 3.2.2 Average ground-level SOA-iv concentrations

Figure 6 depicts the PMCAMx-iv predicted average ground-level concentrations of PM$_{2.5}$ SOA-iv formed from on-road diesel and gasoline IVOC emissions over the modelling domain for May 2008. The highest average PM$_{2.5}$ on-road SOA-iv

concentration is 0.18 μg m$^{-3}$, predicted for Istanbul. The spatial distribution of the average predicted on-road SOA-iv concentrations is similar to that of the new lumped species, with the highest concentrations values located in the corresponding European cities and regions. However, a key difference is that the produced aerosol is transported beyond continental Europe reaching the shores of northern Africa. To quantify the differences between PMCAMx and PMCAMx-iv over the European domain, we estimate the absolute and percentage differences between the average ground-level

concentrations of PM$_{2.5}$ on-road SOA-iv (Figure 7). Compared to PMCAMx, PMCAMx-iv predicts consistently higher concentrations over the domain, with a domain average percentage increase of 60%. The differences between the two models are more prominent over major European cities and over eastern Europe.

The lumped species approach predicts higher concentrations of the IVOC products both in the aerosol and in the gas phase compared to the 1D-VBS approach (Figure S2). Specifically, PMCAMx-iv predicts that in the gas-phase the

secondary products, produced from the oxidation reactions of the new lumped IVOCs from on-road vehicles, have a mean ground-level concentration of 0.019 ppb. On the contrary, the average concentration of the IVOC oxidation products for PMCAMx is predicted to have a value of 0.0008 ppb. For the secondary products in the aerosol phase, PMCAMx predicts a domain averaged concentration of 0.02 μg m$^{-3}$ whereas the corresponding value for PMCAMx-iv is 0.03 μg m$^{-3}$.

The volatility distributions of the on-road SOA-iv products are quite different between the two cases. With the 1D-

VBS approach, 84% of the secondary products in the aerosol phase has a C$^*$ of less than or equal to 1 μg m$^{-3}$, while with the lumped species approach 52% of the secondary aerosol mass has a C$^*$ of 10 μg m$^{-3}$. The mean concentrations of the secondary aerosol products in the 0.1 μg m$^{-3}$ bin are equal between the two models. Furthermore, the average concentration of the aerosol products in the 1 μg m$^{-3}$ bin is higher in PMCAMx than in PMCAMx-iv. A similar shift in volatility is also observed in the gas phase products. Again, in the case of PMCAMx-iv, a significant amount of the gas phase mass has a C$^*$

of 10 μg m$^{-3}$ (46% of the secondary mass in the gas phase), when 76% of the gas phase products of PMCAMx have a C$^*$ greater than or equal to 100 μg m$^{-3}$.



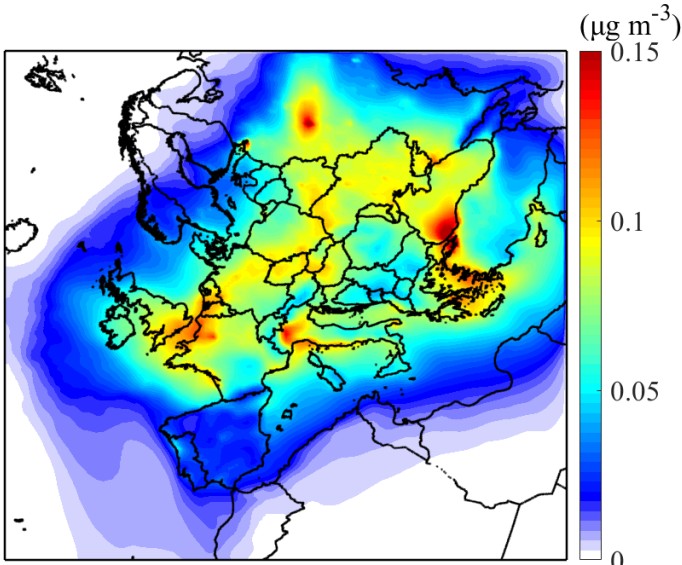

**Figure 6: PMCAMx-iv ("new IVOC scheme" test) average ground level concentrations of on-road transport SOA-iv (μg m⁻³) over Europe for May 2008.**


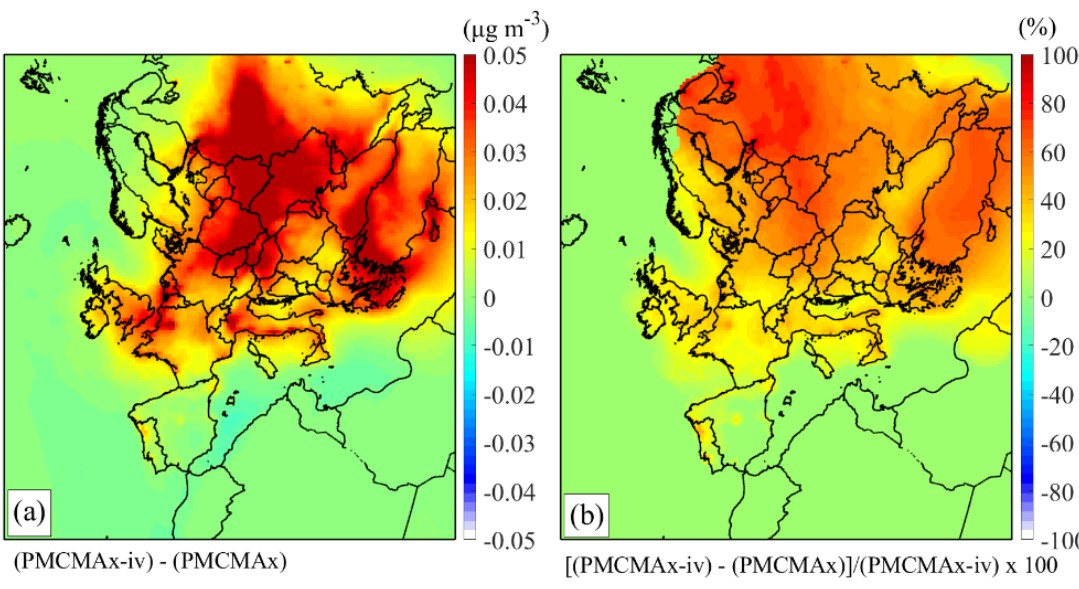

**Figure 7: (a) Absolute and (b) percentage differences between the average ground-level concentrations of PM₂.₅ on-road transport SOA-iv predicted by PMCAMx and PMCAMx-iv.**



### 3.2.3 Contribution of the lumped and individual species to SOA-iv formation

Figure 8 shows the PMCAMx-iv predicted contribution of the seven lumped IVOC species emitted from on-road diesel and gasoline vehicles to the total SOA-iv and to the five SOA-iv volatility bins. The alkanes (ALK6-ALK9) have the highest contribution to the predicted SOA-iv. Specifically, ALK6, ALK7, ALK8 and ALK9 are respectively responsible for the formation of 16%, 36%, 32% and 9% of the SOA-iv, whereas the other species contribute the remaining 7%. Although, ALK6 is the species with the highest emissions and ground-level concentrations, it is only the third most important precursor for SOA-iv according to PMCAMx-iv. This is mainly due to the comparatively lower SOA yields of ALK6. On the other hand, even though ALK7 and ALK8 have lower emissions and predicted ground-level concentrations, because of their elevated mass-based yields they are responsible for the formation of almost 70% of the predicted SOA-iv mass. Moreover, the contribution of ALK8 and ALK9 may be underestimated as their mass-based yields in our SOA scheme are based on the yields of n-heptadecane due to lack of experimental yield data for alkanes with more than 17 carbons.

Analysis of the contribution of individual compounds to SOA-iv formation suggests that the unspeciated cyclic alkanes, which are lumped in the IVOC alkanes, are the most important SOA-iv precursors producing 72% of the total SOA-iv mass, as predicted by PMCAMx-iv (Figure 9). More specifically, unspeciated cyclic alkanes with 15 to 17 carbons (lumped in ALK7) are responsible for 29% of the total SOA-iv and unspeciated cyclic alkanes with 18 to 20 carbons (lumped in ALK8) for another 25% of the total predicted SOA-iv. The contribution of the individual compounds lumped in ALK8 might be a lower bound due to the conservative estimated of their SOA yield parameters. Unspeciated branched alkanes and unspeciated aromatic compounds are respectively the second (14%) and third (5%) most important contributors to the predicted SOA-iv. Regarding the contribution to the secondary product with $C^*$ equal to 10 µg m$^{-3}$, compounds lumped in the lumped PAH species, such as the unspeciated aromatic compounds, naphthalene, and methylnaphthalene isomers, are the second most important contributor.

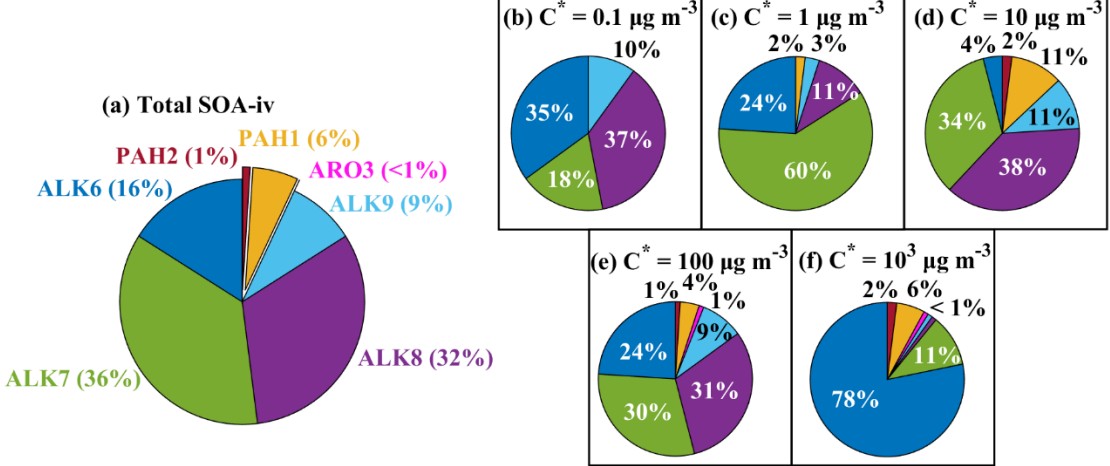

**Figure 8: Contribution of seven lumped species to (a) the total SOA-iv and to the SOA-iv components with $C^*$ of : (b) 0.1, (c) 1, (d) 10, (e) 100 and (f) 10$^3$ µg m$^{-3}$ at 298 K.**



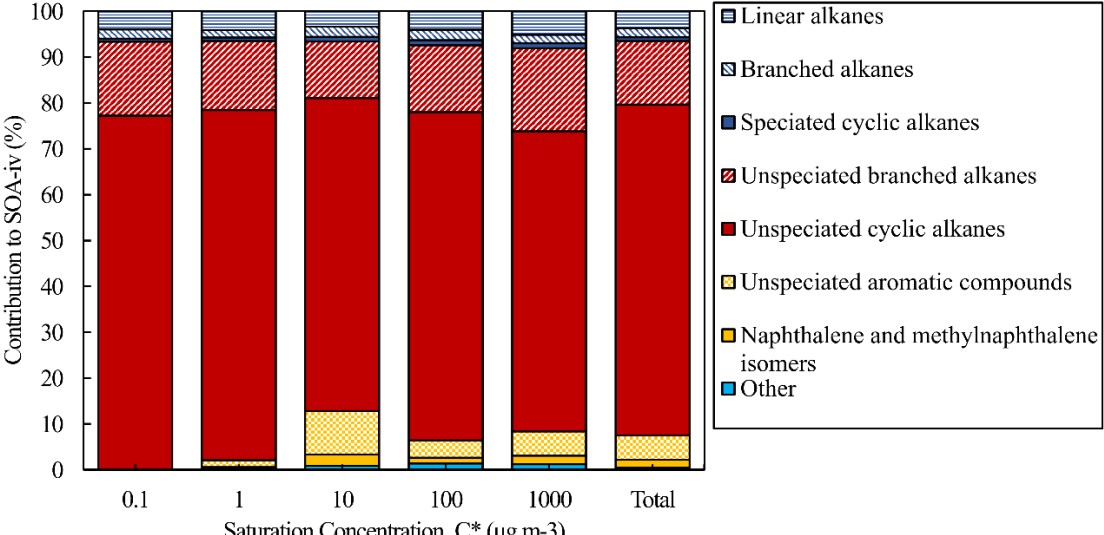

**Figure 9: Average contribution of the individual species to the SOA-iv components for the whole modelling domain.**

### 3.2.4 Contribution of on-road transport IVOCs to SOA and OA levels

According to the results of PMCAMx-iv, the domain averaged contributions of the on-road IVOCs to the SOA and OA are low (less than 10%) (Figure S3). Specifically, using the lumped species approach, the domain averaged contribution of on-road IVOCs to SOA and OA is 3.3% and 1.4%. In European cities, where the on-road diesel and gasoline vehicle emissions are higher, the monthly-averaged contribution of on-road IVOCs to SOA and OA formation is greater than the domain

average. Table S1 includes the contributions of on-road IVOCs to the monthly-averaged SOA and OA concentrations for 44 European cities. In an urban environment, the contribution of on-road IVOCs to the monthly average SOA budget can range from 8.8% (Moscow, Russia) to 2.6% (Oslo, Norway). Moreover, PMCAMx-iv estimates that Minsk (Belarus) is the city with highest contribution of on-road IVOCs to the monthly mean urban OA levels. Across the domain, the on-road SOA-iv that is formed using the lumped species approach contributes significantly more to $PM_{2.5}$ SOA and OA levels compared to

the predictions of the 1D-VBS approach (Figure S3).

Figure S4 depicts the hourly-averaged ground-level SOA and OA concentrations alongside the contribution of on-road IVOCs to the formed aerosol over Moscow for May 2008. On May 12 at 02:00 LT, when the total SOA concentrations peak in Moscow, the contribution of on-road IVOCs to the SOA is 11%. During OA peaks (highest concentration of the month is predicted in May 12 at 08:00 LT), the on-road IVOCs contribute up to 6% of the estimated OA budget. It should be

noted that the maximum contribution of the on-road IVOCs does not coincide with the peak concentrations of SOA or OA. For example, on May 12 at 16:00 LT, after eight hours from the predicted OA peak, on-road IVOCs have a considerable contribution to the estimated OA concentrations (13% of the OA).





### 3.3 Sensitivity Analysis

**3.3.1 The effect of the selected emission factors**

Due to lack of extensive experimental data on IVOCs emitted from European on-road diesel and gasoline vehicles, the lumped IVOC emissions utilized in this study are quite uncertain as they are based on emission factors provided by two experimental studies conducted in the U.S. (Zhao et al., 2015; 2016). Since the percentage of diesel vehicles is higher in Europe compared to the U.S., an "underestimation" is possible as diesel vehicles emit more IVOCs compared to gasoline

vehicles. The "lumped IVOC emissions × 2" sensitivity uses the same model configuration of PMCAMx-iv as the base case, except that the estimated on-road diesel and gasoline vehicle emissions of the lumped IVOC species are doubled.

The effect of the doubled emissions is similar over the whole domain, with the domain-averaged concentrations of the secondary products being two times higher compared to the ones predicted by the "base case" test (Figure S2). In an urban setting, like in the city of Athens (Greece), the highest hourly-averaged on-road SOA-iv concentration predicted by the

"lumped IVOC emissions × 2" sensitivity test is 1.4 μg m$^{-3}$ (14:00 LT on May 28) and accounts for a quarter of the PM$_{2.5}$ SOA mass (Figure S5). For the same hour, the "base case" on-road SOA-iv concentration is 13.6% of the SOA.

### 3.3.2 Effect of the IVOCs chemistry on O$_3$

The second sensitivity test ("no gas-phase chemistry") examines the effect of the more complex IVOCs gas-phase chemistry on ozone production. The yields of the gas-phase products were set to zero and the seven IVOC reactions with the OH

radical produce only SOA-iv species. Figure S6 depicts the absolute and percentage differences between the averaged ground-level concentrations of O$_3$ that are predicted in the "base case" and in the "no gas-phase chemistry" simulations. The absolute differences range from 0 to 0.3 ppb and the percentage differences from 0 to 1.2%. This small effect of on-road IVOCs on O$_3$ production is expected, as the predicted IVOC concentrations are much lower compared to those predicted for VOCs.

Considering that the O$_3$ concentrations vary significantly over time and that the effect of the IVOCs is limited, we focus on the hourly-averaged O$_3$ concentrations over Milan (Italy), an urban area where the effect of IVOCs on O$_3$ appears to be more prominent. More specifically, we focus on the concentrations of O$_3$ over Milan on May 8 as this is the day when the highest difference between the two models is predicted (Figure S7). The contribution of the lumped IVOC species leads to a maximum increase of 1 ppb, which occurs when ground-level ozone concentrations peat at 16:00 LT. It should be noted that

in PMCAMx-iv the yields of the gas-phase products, which contribute to ozone formation, are based on the reactions of VOCs which similar chemical characteristics (ALK5 and ARO2). In the future, as experimental studies provide more information about the IVOC gas-phase chemistry, the model can be updated with more realistic parameters and re-evaluated.





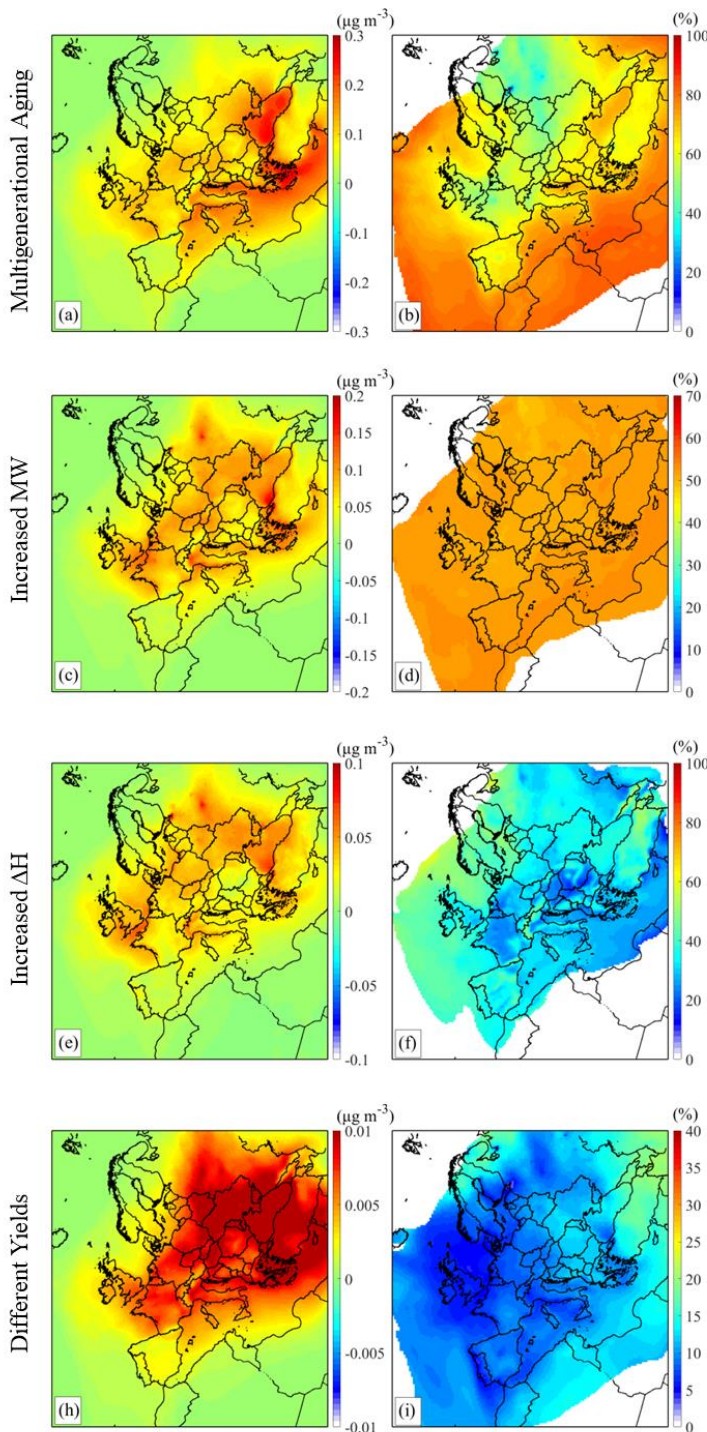

**Figure 10: (a, c, e, h) Absolute and (b, d, f, i) percentage differences between the average ground-level PM$_{2.5}$ concentrations of on-road transport SOA-iv predicted by the "base case" and four sensitivity tests ("multigenerational aging", "MW effect", "ΔH effect" and "different yields".**




### 3.3.3 Multigenerational SOA-iv aging

In the next sensitivity test ("multigenerational aging"), we assess the effect of multigenerational aging by changing the $k_{OH}$ of the corresponding reactions from zero to $4 \times 10^{-11}$ cm$^3$ molecule$^{-1}$ s$^{-1}$. With each aging reaction, the volatility of the condensable products is assumed to decrease by one order of magnitude and there is a 7.5% mass increase to account for the added oxygen.

By allowing the secondary products to further oxidize after they are formed, the mean PM$_{2.5}$ on-road SOA-iv concentrations over the domain increase on average by 67% (Figure 10). The differences between the estimated SOA-iv concentrations of the two tests are higher above the Mediterranean Sea and the Black Sea, where OH levels are high. Moreover, in the "multigenerational aging" test the domain-averaged concentration of SOA-iv is 0.09 μg m$^{-3}$, 3 times higher compared to the one estimated in the "base case". With active multigenerational aging, the produced aerosol becomes less volatile, and more mass ends up in the low volatile bins (Figure S2). This also explains the high differences in remote areas, which are affected by long range transport. Figure S8 compares the predicted diurnal profiles of the on-road transport SOA-iv concentrations over Paris with and without multigenerational aging. After the first peak at 9:00 LT, which is also predicted in the "base case", the "multigenerational aging" test predicts that the concentrations will keep increasing reaching the second peak of the day at 15:00 LT. This high concentration is sustained until sunset when concentrations begin to decrease appropriately. This difference in the diurnal profiles could be useful in determining the importance of the effect of SOA-iv, when measurements become available.

### 3.3.4 The effect of the MW

In the "*MW* effect" sensitivity test of PMCAMx-iv, the molecular weights of the five SOA-iv products were increased from 150 to 250 g mol$^{-1}$. For the simulated period, this leads to a significant increase on the domain-averaged concentration of SOA-iv from 0.03 μg m$^{-3}$ to 0.07 μg m$^{-3}$ (Figure S2). The SOA-iv increase is consistent throughout Europe and it is approximately 50% over the domain (Figure 10). In an urban environment like in the city of Paris, the predicted SOA-iv concentrations also increase by 50%, which leads to a peak value of 1 μg m$^{-3}$ (Figure 11).

By increasing the *MW* of the five SOA-iv products, the molar fraction of each product in the organic aerosol phase decreases, subsequently the equilibrium concentrations of the secondary gas-phase condensable products also decrease leading to increased aerosol phase concentrations. Therefore, this is a relative important parameter for the model that should be better constrained by future experimental studies.





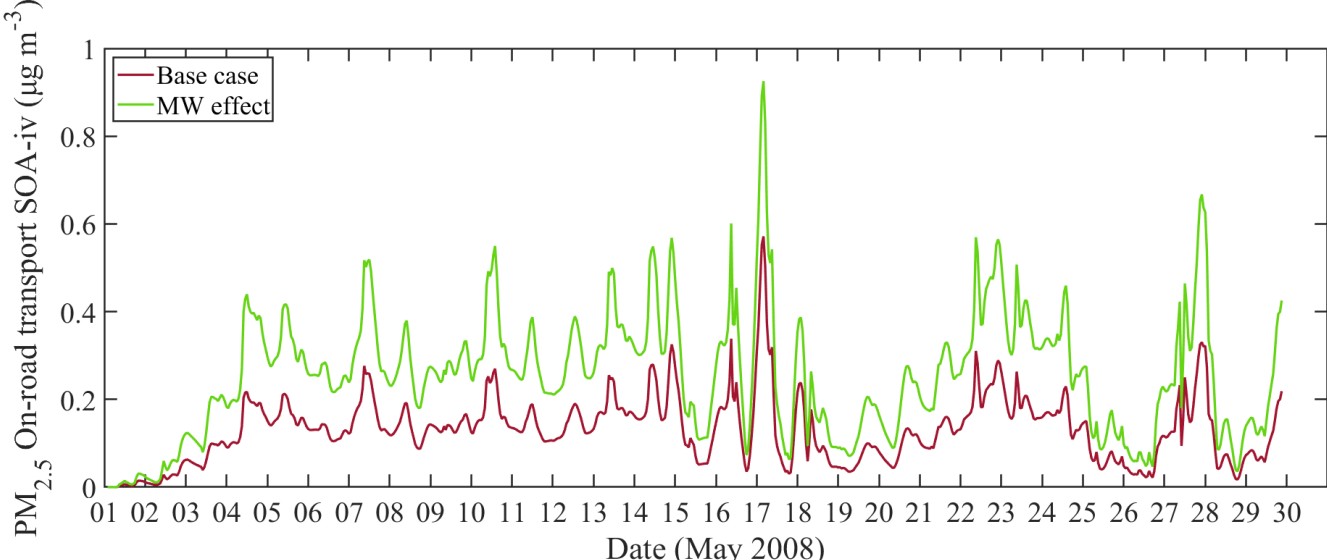

**Figure 11: PM$_{2.5}$ hourly-averaged timeseries for the predicted ground-level concentrations of on-road transport SOA-iv by the base case and the "*MW* effect" test over the city of Paris for May 2008.**

### 3.3.5 The effect of vaporization ΔH

In the "Δ*H* effect" sensitivity test of PMCAMx-iv, the effective vaporization enthalpy was increased from 30 kJ mol$^{-1}$ to 100 kJ mol$^{-1}$. Increasing the Δ$H_{vap}$ results in an increase of the predicted domain average SOA-iv levels by 30%. Compared to the base case simulation, the "Δ*H* effect" sensitivity test leads to higher average concentrations over Northern Italy, the English Channel and Eastern Europe (Figure 10). In Paris, the average daily SOA-iv peak concentration increases to 0.2 µg m$^{-3}$ (Figure S8). Increasing the Δ$H_{vap}$ affects the partitioning of the secondary products and depending on the temperature, this can lead to either an increase or a decrease of the aerosol concentration.

### 3.3.6 SOA-iv yield parametrization

In the "different yields" sensitivity test, we investigate some of the assumptions about the mass-based yields of the new lumped species. Specifically, the mass-based yields of the new alkane species (ALK6-ALK9) under high-NO$_x$ conditions and those of the ARO3 species under both high and low-NO$_x$ conditions are assumed to be two times higher.

On average over the domain, the "different yields" test predicts 10% higher SOA-iv concentrations compared to PMCAMx-iv (Figure 10). The effect is low because of the low concentrations of these compounds. To assess the effect of the increased yields under low-NO$_x$ conditions, we also examine the diurnal profile of the on-road transport SOA-iv concentration over Finokalia (Figure S9). Throughout the day, the "different yields" test predict higher concentrations, reaching up to 0.1 µg m$^{-3}$ at 20:00 LT.





**3.4 Evaluation of the model**

The results of PMCAMx, PMCAMx-iv and the six PMCAMx-iv sensitivity tests are evaluated against ground-level hourly
PM$_1$ OA concentration measurements that were taken in the four EUCAARI measuring sites (Cabauw, Finokalia, Mace
Head, Melpitz). To assess the prediction skills of the models, we consider the normalized mean bias (NMB), the normalized
mean error (NME), the mean bias (MB), the mean absolute gross error (MAGE), the fractional bias (FBIAS), and the
fractional error (FERROR). The definitions and estimated values of these metrics are presented respectively in the
Supplement S1 and in Table S2.

Overall, neither PMCAMx-iv nor any of the sensitivity tests have any significant effect on the performance of the
model (Table S2). These results are expected, as the four measuring stations are located in either remote or rural areas, where
the predominant source of OA is long range transport. In almost all simulations cases the contribution of the on-road IVOCs
to the PM$_1$ OA concentrations in the four measuring sites is less than 5% (Table S3). The only exception is the
"multigenerational aging" test, which predicts that the contribution of on-road IVOCs to the PM$_1$ OA levels of Finokalia can
be up to 10%. Even in that case, the change in the mean predicted OA concentration is only 0.1 μg m$^{-3}$ (Table S2). The
evaluation of the model against detailed OA composition measurements in urban areas will be the focus of future work.

**4. Conclusions**

A lumped species approach for the simulation of IVOCs and SOA-iv formation was implemented in a CTM, PMCAMx-iv.
The new model includes updated on-road transport emissions, describes more realistically IVOC chemistry and utilizes a
new method for simulating aerosol formation from IVOCs.

        For the simulated period, the maximum ground-level SOA-iv concentrations predicted by PMCAMx-iv, over major
European cities like London, Paris, and Athens, can reach up to 0.63 μg m$^{-3}$. If multigenerational aging is considered, the
predicted on-road SOA-iv concentrations increase on-average over the European domain by 67%. This means that in an
urban setting like the city of Athens, on-road IVOCs can account for 30% of the total SOA. Moreover, PMCAMx-iv predicts
that among the simulated IVOCs, the most important contributors to SOA-iv formed from on-road diesel and gasoline
vehicles are unspeciated cyclic alkanes, contributing 72% of the total SOA-iv mass. Especially unspeciated cyclic alkanes
with 15 to 20 number of carbons are estimated to be the most important SOA-iv precursors. Compared to a previous version
of the model, that utilizes the 1D-VBS approach to simulate IVOCs, the domain averaged concentrations of on-road SOA-iv
have increased by 60% over the simulated domain. Also, with the implementation of experimental SOA-iv yield data in
PMCAMx-iv, the formation of on-road SOA-iv occurs earlier in the day than the 1D-VBS approach.

        The results presented in this work may be a lower bound as the SOA-iv yields of these compounds are based on
rough approximations. The sensitivity analysis on the yields and the SOA-iv parameters showed that the molecular weight
can have a significant impact on the predicted SOA-iv concentrations. Future experimental results on those aspects will help
constrain the effect of the assumptions made. Also due to lack of experimental data, the current model only simulated the



OH oxidation reaction of IVOCs. Further investigation of the oxidation reactions of IVOCs during night-time is needed. The simulations indicate that as expected on-road IVOCs and their effect on SOA and OA levels over Europe are more prominent locally in urban environments.

## 5. Supplementary information

## 6. Author contribution

SEM and SNP designed the research. SEM implemented the lumping scheme to the model, performed the simulations and analysed the results. SEM wrote the paper with input from SNP.

## 7. Competing interests

The authors declare that they have no conflict of interest.

## 8. Funding

This work has received funding from the European Union's Horizon 2020 research and innovation program under project FORCeS, grant agreement no. 821205 and from the European Union's Horizon Europe (2021-2027) research and innovation program under project EASVOLEE, grant agreement no. 101095457.

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
