# Peer review of "Contribution of intermediate volatility organic compounds from onroad transport to secondary organic aerosol levels in Europe"

_EGUsphere, 2023_

## Author Response (AR1)

**Responses to the Community Comment #1 of Zeyu Liu**

**(1)** Line 31. It might me intermediated volatility organic compounds (IVOCs).
We have corrected the typo.

**(2)** Line 32 I don't think these three citations include the aircraft works, please check.
We have included the appropriate citation for aircraft engines (Presto et al., 2011).

**(3)** Line 51 There should be more data about the UCM percentage of total IVOCs in different sources. This will give readers an intuitive feeling about the importance of UCM. e.g. on-road vehicles (Zhao, 10.1021/acs.est.5b02841), ships (Liu, 10.1021/acs.est.2c03589), construction machinery (Qi, 10.1021/acs.est.9b01316), solid fuels (Qian, 10.1021/acs.est. 0c07908).
Following the suggestion of the commentator, we have updated the corresponding section so that the reader can understand the importance of the UCM for different sources.

**(4)** Line 56 Actually the eleven bins were calculated by the retention times of thirteen n-alkanes (C11-C23).
As the commentator correctly indicates, Zhao et al. (2014) determined the start and end of the retention time for the nth bin by utilizing the retention times of the $C_n$, $C_{n-1}$ and $C_{n+1}$ n-alkanes. Therefore, to create 11 bins which correspond to the 11 n-alkanes that span the range from dodecane ($C_{12}H_{26}$) to docosane ($C_{22}H_{46}$), additional information about the retention times of undecane and tricosane is needed. To avoid confusion and since the analysis of the details of the method of Zhao et al. (2014) is beyond the scope of our work, we have rephrased the corresponding sentences to avoid unnecessary confusion.

**Responses to the Comments of Referee #1 (Paolo Giani)**

**General Comments:**

**(1)** Manavi and Pandis present new PMCAMx simulations with their novel scheme to compute SOA from gasoline- and diesel-emitted IVOCs oxidation. Overall, I found the article well-written. The literature review in the introduction is definitely comprehensive and gives a great overview of recent and relevant SOA work. The description of the numerical experiments reads well and results are presented coherently. Although the findings of this work are not necessarily unexpected, I feel that this kind of analysis can be useful as a future technical reference for practitioners interested in incorporating the latest advances in SOA modeling and understanding what their effects will be on the overall model predictions. With that said, I do have a few suggestions that could likely improve the usefulness of the manuscript for ACP readers, which I detail here below. Specifically, the manuscript could probably benefit from a separate discussion section, where some of the discussion points I think it would be worth mentioning are outlined below. There are a few technical corrections as well that I took note of while reading the manuscript.

We appreciate the positive assessment of our work by the referee. Following his comments and recommendations, we have revised the manuscript to provide additional information about the application of the lumped species approach to IVOCs emitted from on-road diesel and gasoline vehicles, the sensitivity tests, and the evaluation of PMCAMx-iv. Our responses (in regular font) follow each comment (in italics).

**Specific comments**

**(2) Model added complexity**. If I understand correctly, the new scheme tracks 7 IVOC species compared to 4 surrogate volatility-based species in the classic 1D VBS approach and simulates 14 reactions in total (7 in gas-phase and 7 for low-volatile SOA that can partition between gas/aerosol phase). I wonder if the authors feel that the results presented in the manuscript will justify the added complexity of the model, looking into the future. Examples of this added complexity include: (i) additional RAM memory where the new 3D variables need to be stored (ii) additional computational time to compute the reactions (although I'd suspect that this is not a large increase), (iii) additional free parameters (e.g. reaction rates, DHs, …) that need to be constrained, (iv) additional efforts by practitioners to speciate their emission inventories coherently with the new species, if they decide to use the new version of SAPRC, (v) additional efforts and resources to test and validate the model, such as this manuscript. In other words, given that the total OA concentrations are rather insensitive to gasoline- and diesel-emitted IVOCs oxidation (as shown in this manuscript), I wonder if the authors have a sense if we as a community should spend our (finite!) resources in better constraining some of the uncertain parameters presented in this work and refine the proposed scheme, or focus more on some other aspects of OA modeling which CTMs might be more sensitive to. Maybe a discussion of the added complexity of the model (possibly touching upon some of the aspects that I have listed before – how much more RAM? How much more computational time? How many new parameters? How practical would it be for operational users to incorporate the new speciation?) would be beneficial to see the broader perspective where this work fits in.

Following the recommendations of the reviewer, we have revised the manuscript to better present the purpose of introducing more complexity when simulating the gas-phase chemistry and SOA formation of IVOCs. Compared to the 1D-VBS approach, which describes the compounds only based on their effective saturation concentrations, our model provides information about the chemical characteristics of the SOA-iv precursors. Our aim is to contribute towards identifying which of the thousands of compounds in the IVOC range are the most important SOA precursors and provide insights about which parameters should be better constrained in the future. For example, regarding the IVOCs emitted from on-road vehicles, we have shown that cyclic alkanes with 15 to 20 number of carbons are potentially the most important SOA-iv precursors. However, smog chamber literature has only focused on specific linear alkanes or PAHs. Despite the added complexity of PMCAMx-iv, the required CPU time increases by only 2% compared to PMCAMx. The gas-phase chemistry module, to which most of the complexity is added, is not computationally intensive. It is worth noting that the simulation of SOA-iv formation with the 1D-VBS approach requires 22 species (4 primary IVOCs and 9 pairs of gas/aerosol secondary species), 12 gas-phase reactions with the OH radical and 9 partitioning reactions. With the lumped species approach, we simulate 17 species (7 lumped species in the gas-phase and 5 pairs of gas/aerosol secondary species), 7 gas-phase reactions with the OH radical (or 12 in the case of active multigenerational aging) and 5 partitioning reactions. In PMCAMx-iv we currently use both schemes, because the lumped species are used to simulate only one source of IVOCs, that of on-road diesel and gasoline vehicles. If one simulates all IVOC sources with the new scheme, the 1D-VBS species and the corresponding reactions that describe SOA-iv formation could be removed from the model and therefore the model would become faster. Regarding the implementation of the lumped species scheme to other CTMs or global models, all the SOA parametrizations can be taken directly from Manavi and Pandis (2022). If the model utilizes the SAPRC gas-phase mechanism, then the implementation is straightforward. SAPRC-22 (the latest version of the mechanism) (Carter et al., 2023) includes the ALK6 species and a species to explicitly represent naphthalene, but it does not include any of the higher alkane species of the more complex PAH species. Further details about the implementation of the lumped species approach to other models can be found in Manavi and Pandis (2022). A brief discussion of these interesting points has been added to the revised manuscript.

**(3) Why Gasoline and Diesel Emissions?** A follow-up point to the first specific comment – Can the authors comment on the decision of focusing on gasoline and diesel IVOC emissions, when other evidence points to biomass burning being the main emission source for IVOCs? For instance, Basla et al. (2022, Atmosphere); Giani et al., 2019 (Atm Env) estimated that biomass burning IVOC emissions contribute to more than 90% of the total IVOC emissions in Northern Italy during wintertime, whereas gasoline and diesel emissions only account for 1% and 5% (using the updated Zhao et al. scaling factors). Accordingly, they showed that revising biomass burning IVOC emissions makes a large difference for SOA production, but not so much with gasoline and diesel vehicles emissions (since their contribution is so small). In summertime, biomass burning emissions are lower but still make up the largest fraction. I understand that detailed experiments were performed on gasoline and diesel vehicles by Zhao

et al., but I wonder if there are any other reasons why the authors decided to embark on this task and if those reasons could be laid out in the discussion section.

The simulation of a source of IVOCs with the lumped species approach requires information about the chemical characteristics of the emitted IVOCs. Due to various experimental limitations, identifying the chemical species in the IVOC range remains challenging. Zhao et al. (2014; 2015; 2016) were one of the first studies in the field that constrained the UCM and provided information about its chemical characteristics. Therefore, we have chosen the work of Zhao et al. (2015; 2016) as the starting point for the first application of our method. The reviewer correctly indicates biomass burning can be an important source of SOA both during the winter, due to residential wood combustion, and during the summer, due to wildfires and agricultural waste burning. IVOCs emitted by biomass burning are currently simulated in our model with the 1D-VBS. Using PMCAMx and the 1D-VBS scheme, Theodoritsi and Pandis (2019) have evaluated the contribution of IVOCs to biomass burning OA. As more experimental studies provide more detailed chemical emission profiles for other sources of IVOCs, we will be able to incorporate their simulation to PMCAMx-iv. Specifically, biomass burning will be the focus of future work. We have made the necessary changes in Section 2.3.1, so the reader can more clearly understand why we choose on-road diesel and gasoline vehicles as a starting point for our simulations.

**(4) Model Validation.** The authors honestly show that the model performance does not change much with PMCAMX-iv compared to PMCAMx, as expected because of the comparison with rural sites. I have two suggestions/concerns on this point. First, why did the authors choose to simulate May 2008 for a comparison with EUCAARI remote sites then, if they expected that no sensitivity would be observed? Could they have considered any urban site that measured PM1 OA? Second, there are AMS/ACSM measurements that have been further apportioned with PMF in different components, including a Hydrocarbon-Like OA (HOA) component – e.g., Bressi et al. (2016, ACP). HOA could probably be a good testbed for SOA derived from gasoline and diesel emissions, and that could be an interesting way of directly evaluating the SOA-iv calculations. Have the authors considered this possibility?

We recognize that the evaluation of PMCAMx-iv in this work is necessarily limited, because of the lack of the corresponding measurements in past field campaigns. To directly evaluate the performance and reliability of the lumped species approach, measurements would be needed for both the gas-phase concentrations of IVOCs and SOA-iv. For the evaluation of our model for on-road transportation, these measurements would be even more challenging because they would require estimation of the SOA-iv from this specific source. To the best of our knowledge no such information exists right now. As the reviewer suggests, AMS measurements and PMF analysis could be in principle used for the evaluation of our model. However, the available HOA concentrations should not include most of the SOA-iv because it is similar to oxidized OA based on past studies (Chen et al., 2016). If SOA-iv is part of the OOA factors (LO-OOA and MO-OOA) determined by previous AMS studies in urban areas, its deconvolution from the other OOA components is currently practically impossible. We hope that our study will provide motivation for the development of methods for the determination of SOA-iv in at least polluted urban areas. This is now discussed in the revised manuscript.

**(5) Gas Phase Chemistry Importance.** In section 3.3.2, the authors show that the new IVOC scheme does not seem to affect the gas-phase chemistry and $O_3$ concentrations much. Based on these results, would concluding that we probably don't need this added complexity in gas-phase chemistry (going back to point 1) be fair? In other words, do the authors have any evidence that we should definitely keep these added gas-phase reactions in SAPRC, based on their data?

Even if our current results suggest that on-road transportation IVOCs have a small effect on regional $O_3$ concentrations over Europe during the simulated period, we believe that these results should not be generalized for different areas of the world or for the simulation of other sources. For example, Li et al. (2011) have shown that the IVOC chemistry could play a non-negligible role in polluted urban environments, like that of Mexico City. In another study, focusing on IVOCs emitted from consumer products Li et al. (2018) suggested that at least some IVOCs can suppress ozone formation. Given that the addition of the lumped IVOCs to the gas-phase chemistry is not computationally expensive, we suggest that the gas-phase reactions are kept in the lumped species schemes and that their importance should be further tested in different conditions and scenarios. Section 3.3.2 has been updated to include the appropriate references that suggest that IVOCs could affect ozone formation under conditions different from those in the present study.

**(6) Seasonality.** Would any of the results presented in this work change if simulations were done in wintertime? Would the authors expect more or less sensitivity to the new mechanism? Could that be speculated in the discussion section? This would probably give a sense to readers on what to expect (although not backed up by simulations yet) if met conditions were to change. This is an excellent question and a nice topic for future work. However, there are several factors involved and the answer is not clear. For example, previous work has suggested that PMCAMx (and similar models) has additional difficulties in reproducing oxidized OA levels during the winter in parts of Europe (Fountoukis et al., 2014). However, this could be due to several reasons including missing emissions, difficulties with meteorology, missing processes (like nighttime processing of biomass burning emissions, etc.). One expects that the transportation IVOC concentrations will tend to increase (due to weaker mixing), but photochemistry will slow down. On the other hand, the lower temperatures will favor partitioning of the SOA-iv components towards the particulate phase. Given all these issues, we would prefer to avoid speculating but to instead underline in the revised paper the need to study these effects during winter.

**(7) Emissions/Concentrations Correlation.** If I look at Figure 1 from the authors' previous article (Manavi and Pandis, 2022) and Figure 1 in the present manuscript, I do see some good similarities between the IVOC emissions (in the previous paper) and concentrations (in this paper). Is it fair to say that IVOC concentrations scale pretty well and linearly with IVOC emissions? In other words, can the authors estimate a correlation coefficient and possibly show a regression line between the two? This would give a back-of-the-envelope estimate on how much the IVOC concentrations are expected to increase with an increase in emissions. Another way of checking this would be to plot the Paris emission diurnal profile and compare it with Figure 2. Does that have the same shape?

This is an interesting topic that has a temporal and a spatial dimension. Following the recommendation of the reviewer we have added an extra figure in the supplementary material section that shows the average diurnal profiles of total IVOC emissions from on-road diesel and gasoline vehicles over Paris and the corresponding IVOC concentrations. Since the IVOCs undergo several processes in the model, such as dilution and reactions with the hydroxyl radical, the diurnal profile of their emissions rates is different than this of their concentrations. However, during the morning the two profiles are relatively similar. For the spatial dimension, for which the temporal differences are averaged out, the situation is simpler and there is good correlation. This is discussed in the sensitivity analysis of the paper in which we show that the doubling of the emissions leads to an approximate doubling of the predicted concentration. A brief discussion of these points has been added.

**(8) Multigenerational aging.** In section 3.3.3, the authors show that SOA-iv is rather sensitive to further oxidation in the atmosphere. Do the authors have a sense if this is realistic or not, based on existing literature? In other words, if I am willing to use the new scheme, would the authors recommend turning multigenerational aging on in the model? Bressi et al. (2016, ACP) seem to show that significant contributions of aged secondary organic aerosols are observed throughout the year. Would that be some evidence in favor of letting SOA further oxidate? Or is there not enough evidence to claim that it would be more realistic to turn it on rather than off?

As the reviewer points out, several studies have shown that ambient OA is dominated by its oxygenated component in both rural and urban sites. The high O:C ratio that is measured in ambient studies suggests that the OA is predominantly SOA that is the product of multiple generation oxidation steps in the atmosphere. Similarly, smog-chamber experimental studies have shown that the SOA produced from the oxidation of PAHs in the IVOC range can have an O:C ratio close to 0.7 (Chen et al., 2016). This underlines the importance of aging processes. In our model, PAHs are the third most important SOA-iv precursor class, contributing 5% of the mass. Unfortunately, the smog chamber experimental data from alkanes in the IVOC range are more complicated. For example, smog chamber experiments conducted with linear and branched alkanes in the IVOC range have shown that the O:C ratio of the produced SOA has an O:C ratio below 0.3, meaning that the produced aerosol in these studies is more closely related to first or at least early generation products (Tkacik et al., 2012; Yee et al., 2013). Yee et al. (2013) investigated the oxidation of hexylcyclohexane (cyclic alkane with 16 carbons) and showed that the corresponding SOA has an O:C above 0.3, indicating that the oxidation products of these compounds may be susceptible to further oxygenation. In our model, more than half of the produced SOA-iv is attributed to the oxidation of unspeciated cyclic alkanes with 15 to 20 carbons. To conclude, in the current literature, there is evidence that including the multigenerational aging of SOA-iv in our model would better reflect ambient concentrations and the oxidation state of SOA. However, more laboratory and ambient studies are needed to provide a definite answer. For these reasons, we have chosen to add the appropriate multigenerational reactions to our model but kept them as an option for the discretion of the user. We have updated Section 3.3.3 in the paper to better reflect all the points made above.

**(9) Line 285-288: Early-Morning SOA.** How much do you honestly think that this new scheme could improve Tsimpidi's observation that SOA increase during morning hours is underestimated? The peak difference with the baseline in Figure 5 is about 0.02 µg/m$^3$ (over an urban area). To me, it doesn't seem nearly large enough to explain that underestimation (even if emissions were even higher than what estimated in this manuscript).

This a good point that needs clarification. It is true that the new model, which includes the updated scheme only for only IVOCs emitted from on-road diesel and gasoline vehicles, cannot fully close the gap between the measured and predicted morning SOA concentrations in Mexico City. The comparison with the results of Tsimpidi et al. (2011) concerns the diurnal profile of the predicted SOA-iv concentrations, rather than the absolute values. Tsimpidi et al. (2011; 2014) underpredicted the morning peak and overpredicted the peak later in the afternoon, suggesting that the chemistry of SOA formation (from all sources and not just transportation) was slower compared to what is happening in the atmosphere. This is now clarified in the paper.

**(10) Sensitivity analysis.** I feel that the sensitivity analysis part of the manuscript can be improved. Maybe the authors can choose a couple of variables of interest (e.g., max hourly SOA, domain-averaged O$_3$ concentrations) and show a summary comparison figure with their values in all the different experiments, just to have visual guidance on what are the most relevant sensitivity factors. Also, when reporting the sensitivity values (e.g. line 385) it would be useful to have the reference values as well, to understand how large the difference is. Same applies for the other sensitivity sections.

We have followed the recommendations of the reviewer and revised the sensitivity section. A figure with the maximum hourly SOA-iv concentrations over Paris has been added. Similarly, we have updated the individual sections that describe the different sensitivity tests to include the reference values, so that the effect of each test would be more apparent to the reader.

**Technical corrections**

**(11)** Line 78: I believe 'fist' is a typo.
We have corrected the typo.

**(12)** Line 118: Remove comma after although.
The comma has been removed.

**(13)** Line 399: I believe 'peat' is a typo.
We have corrected the typo.

**References**

Basla et al., 2022 Simulations of Organic Aerosol with CAMx over the Po Valley during the Summer Season. Atmosphere.
Bressi et al., 2016 Variations in the chemical composition of the submicron aerosol and in the sources of the organic fraction at a regional background site of the Po Valley (Italy). ACP.

Carter, W. P. L. : Documentation of the SAPRC-22 mechanism, https://intra.engr.ucr.edu/~carter/SAPRC/22/S22doc.pdf. last acess: 17 Nov 2023, 2023.

Chen, C. L., Kacarab, M., Tang, P., and Cocker, D. R.: SOA formation from naphthalene, 1-methylnaphthalene, and 2-methylnaphthalene photooxidation, Atmos. Environ., 131, 424-433, https://doi.org/10.1016/j.atmosenv.2016.02.007, 2016.

Fountoukis, C., Megaritis, A. G., Skyllakou, K., Charalampidis, P. E., Pilinis, C., Denier van der Gon, H. A. C., Crippa, M., Canonaco, F., Mohr, C., Prévôt, A. S. H., Allan, J. D., Poulain, L., Petäjä, T., Tiitta, P., Carbone, S., Kiendler-Scharr, A., Nemitz, E., O'Dowd, C., Swietlicki, E., and Pandis, S. N.: Organic aerosol concentration and composition over Europe: insights from comparison of regional model predictions with aerosol mass spectrometer factor analysis, Atmos. Chem. Phys., 14, 9061-9076, 2014.

Li, G., Zavala, M., Lei, W., Tsimpidi, A. P., Karydis, V. A., Pandis, S. N., Canagaratna, M. R., and Molina, L. T.: Simulations of organic aerosol concentrations in Mexico City using the WRF-CHEM model during the MCMA-2006/MILAGRO campaign, Atmos. Chem. Phys., 11, 3789–3809, https://doi.org/10.5194/acp-11-3789-2011, 2011.

Li, W., Li, L., Chen, C.-l., Kacarab, M., Peng, W., Price, D., Xu, J., and Cocker, D. R.: Potential of select intermediate-volatility organic compounds and consumer products for secondary organic aerosol and ozone formation under relevant urban conditions, Atmos. Environ, 178, 109-117, https://doi.org/10.1016/j.atmosenv.2017.12.019, 2018.

Tkacik, D. S., Presto, A. A., Donahue, N. M., and Robinson, A. L.: Secondary organic aerosol formation from intermediate-volatility organic compounds: cyclic, linear, and branched alkanes, Environ. Sci. Technol., 46, 8773-8781, https://doi.org/10.1021/es301112c, 2012.

Yee, L. D., Craven, J. S., Loza, C. L., Schilling, K. A., Ng, N. L., Canagaratna, M. R., Ziemann, P. J., Flagan, R. C., and Seinfeld, J. H.: Effect of chemical structure on secondary organic aerosol formation from C12 alkanes, Atmos. Chem. Phys., 13, 11121–11140, https://doi.org/10.5194/acp-13-11121-2013, 2013.

Zhao, Y., Hennigan, C. J., May, A. A., Tkacik, D. S., de Gouw, J. A., Gilman, J. B., Kuster, W. C., Borbon, A., and Robinson, A. L.: Intermediate-volatility organic compounds: a large source of secondary organic aerosol, Environ. Sci. Technol., 48, 13743-13750, https://doi.org/10.1021/es5035188, 2014.

Zhao, Y., Nguyen, N. T., Presto, A. A., Hennigan, C. J., May, A. A., and Robinson, A. L.: 715 Intermediate volatility organic compound emissions from on-road diesel vehicles: chemical composition, emission factors, and estimated secondary organic aerosol production, Environ. Sci. Technol., 49, 11516-11526, 2015.

Zhao, Y., Nguyen, N. T., Presto, A. A., Hennigan, C. J., May, A. A., and Robinson, A. L.: Intermediate volatility organic compound emissions from on-road gasoline vehicles and small off-road gasoline engines, Environ. Sci. Technol., 50, 4554-4563, 2016.

**Responses to the Comments of Referee #2**

**General comment**

**(1)** The paper contributed by Manavi and Pandis applied a newly developed model (with more details found in a prior study Manavi and Pandis, GMD, 2022) to simulate SOA formed from IVOCs emitted by diesel and gasoline vehicles in Europe. A few sensitivity tests were conducted to investigate the associated impacts on SOA simulations. Generally, the paper is well-written and informative to readers who are interested in SOA modeling with CTMs. However, I have some concerns about the contents of the paper, which I think should be addressed before it is published.

We do appreciate the positive assessment of our manuscript. We have made several changes to the paper to improve it following the recommendations of the reviewer. These changes are described below (in regular font) following each comment of the reviewer (in italics).

**Specific comments**

**(2)** Since this work is an application of the new model, I feel model evaluation is an important part, which can convince readers the simulations are reliable. As such, I would suggest putting the evaluation forward, probably, Section 3.1, and presenting comparisons of modeled and measured OA in the form of time series, diurnal variations, etc.

We have followed the recommendation of the reviewer and moved the evaluation of the model to Section 3.3. Moreover, we have updated the supplementary material to include a figure depicting the timeseries of the measured and predicted $PM_1$ OA concentrations for the simulated month and a figure depicting the diurnal profile of the predicted and measured concentrations.

**(3)** It might be frustrating to see that the model performance is not sensitive to any tests (Table S2), making the tests less meaningful. The authors have pointed out that this might be due to the observational sites being located in remote or rural areas. It is possible to select a campaign at a site that is substantially affected by on-road emissions?

We understand that the comparison is not ideal but unfortunately, the focus of the EUCAARI campaign was on the concentrations and composition of background PM over Europe. For the selected period, continuous PM measurements exist only from rural background stations. Please also see our response to Comment 4 of Reviewer 1, in which the evaluation of the model predictions for SOA-iv are discussed.

**(4)** At present, the chemistry for IVOC is highly uncertain. Readers would be very interested in how IVOCs are reacted in the model, what are the products (volatile products that are involved in chemistry), and the stoichiometric coefficients. It would be good to provide some information even if they have been depicted in the Manavi and Pandis, GMD, 2022.

This is a good suggestion. We have updated the supplementary material, and it now includes a short description of the gas-phase chemistry and SOA parametrization of the IVOCs that are included in the model.

**(5)** What are the motivations for the tests of MW effect, H? It would be good to add some context.

Only a small fraction of the simulated IVOCs has been studied in smog chambers. As a result, there is significant uncertainty regarding the required parameters for the model. We had to use reasonable guesses for some of these parameters during the development of our lumping scheme. With the sensitivity tests, we wanted to evaluate the effect of the assumed values so that future work can focus on the most important ones. The molecular weight of the produced SOA is a good example and is a parameter that receives little attention in both past laboratory and modeling studies. The effective vaporization enthalpy has received more attention but is also a good example of an uncertain model parameter. We have added explanations in the respective sections of the manuscript to provide more context for our changes.

**(6)** Line 152: IVOCs from other sources are estimated using 1D-VBS approach. Do the authors use POA and IVOC-to-POA ratios to estimate IVOC emissions? This might underestimate IVOC emissions from VCPs (or solvent use), which is a large source of IVOCs while not emitting POA.

Besides on-road diesel and gasoline vehicles, the rest of the IVOC emissions are based on the 1.5 IVOC-to-POA scaling factor. As the reviewer correctly indicates, the literature suggests that use of this scaling factor may not be appropriate for all sources (Lu et al., 2018). The goal of our study is to first test the application of the new lumping IVOC scheme to one source, so that in the future we will be able to include the representation of more sources. A brief discussion of this point has been added to the paper.

**References**

Lu, Q., Zhao, Y., and Robinson, A. L.: Comprehensive organic emission profiles for gasoline, diesel, and gas-turbine engines including intermediate and semi-volatile organic compound emissions, Atmos. Chem. Phys., 18, 17637-17654, 2018.

Responses to the Comments of Referee #3

**(1)** The work presented here is a follow-on from a recent paper by the same authors in GMD in which they describe the changes made to the PMCAMx model to include: firstly, additional emissions of IVOCs from diesel and gasoline road transport, categorised into 7 different types of lumped species; and, secondly, the SOA-forming chemistry of these lumped species. In the current manuscript they apply this enhanced model to the simulation of atmospheric composition, particularly the SOA component of PM2.5, over Europe during May 2008. This simulation period coincides with the EUCAARI intensive measurement campaign. The authors compare the simulated SOA surface concentrations of this new model with the original model, and also undertake 6 sensitivity experiments in which they individually alter aspects such as the total IVOC emissions from road transport, the extent of multigenerational aging of the condensable products, and the yields of SOA under low-$NO_x$ and high-$NO_x$ conditions.

Our responses (in regular font) follow each comment of the reviewer (in italics).

**General comments**

**(2)** This is an important area of PM2.5 air quality science. In their earlier GMD paper, the authors present a sensible approach to improving the sophistication of handling simulation of IVOC-related SOA formation. The work presented in the current manuscript is straightforward in concept but has been methodically undertaken and is clearly described and visualised. The Introduction provides a well-written description of the current state-of-the-science and clear rationalisation for the work. I support its publication in ACP after some revision. It should be noted, however, that the current manuscript does need to be read in conjunction with the GMD paper, if the reader is not already fully familiar with the changes made to the model to improve the modelling of IVOC-derived SOA. The authors include rather little discussion of their work. Here are two areas in which further comment would be helpful.

We appreciate the positive feedback of the referee. Following the corresponding suggestions, we have enhanced the discussion section of our manuscript so that our results are presented to the reader more coherently and together with other recent advancements of our understanding of IVOCs and SOA-iv.

**(3)** First, a discussion of where next with their model parameterisation. The authors describe the results of their sensitivity tests but provide little opinion of whether they believe any (or which) of the sensitivity tests may better reflect reality or what further modifications should be made to enhance the representation and parameterisation of IVOCs and their atmospheric chemistry.

Currently, due to lack of adequate experimental data, there is still uncertainty regarding certain aspects of the chemistry of IVOCs in the atmosphere. For example, depending on the chemical class of the IVOCs there might be stronger or weaker competing effects of fragmentation and functionalization. This will have a direct effect on certain parameters of the formed SOA, such as the aerosol yields, the oxidation state, and the molecular weight. Without the appropriate smog chamber experiments or without direct ambient measurements, it is difficult to provide a definite answer on whether any of the sensitivity tests reflect better the real atmospheric

11

conditions. Following the recommendation of the reviewer we have enhanced the discussion of our sensitivity tests to include comparisons with other experimental and modelling studies. Further future potential modifications of the model include expanding it to represent more IVOC sources, updating the gas-phase chemistry model to better reflect IVOC chemistry and updating the aerosol yields and parameters.

**(4)** Secondly, the simulation period for the VOC and IVOC emissions is May 2008. This is 15 years ago and road transport emissions of (I)VOCs will now be considerably lower, and hence the impact of road transport emissions on SOA and ozone formation will also now be less than presented in this paper. There is also likely a different split between diesel and gasoline on-road vehicles in the present-day compared with 2008. What is the authors' view on whether road transport IVOC-related SOA formation is a current-day issue in Europe? Connected with this comment, the title of the manuscript implies the work is an evaluation of current contribution of IVOC to SOA, whereas in reality it is an evaluation of the contribution of IVOC to SOA 15 years ago. The authors could think of an alternative title.

Over the past decade, the imposed EU regulations on on-road vehicles have resulted in a reduction of ambient VOC concentrations. Given that there is a strong correlation between VOC and IVOC emissions from on-road diesel and gasoline vehicles, we expect that the current-day IVOC concentrations will be lower compared to the ones predicted by our model for the EUCAARI intensive. Nevertheless, the estimation of our emissions was based on two studies conducted in the U.S. (Zhao et al., 2015; 2016) and this introduces significant uncertainties. Without measurements of IVOCs emitted from European vehicles it is difficult to estimate the true effect of the imposed regulations. For example, Fang et al. (2021) showed that in China the emission factors of IVOCs emitted from on-road diesel and gasoline vehicles are higher compared to those in the studies of Zhao et al. (2015; 2016). There are ongoing EU projects focusing on the measurement of IVOC emissions in Europe and we hope that we will be soon able to estimate their current contribution to the SOA in Europe and address the important question raised by the reviewer. Our title does not imply that the paper addresses the current contribution, so we think that it is not misleading, and we would prefer to keep it in its present general form. A brief discussion about changes in the last decade has been added to the revised paper.

**(5)** L114-115: This sentence needs rephrasing so that the scientific sense is the other way around. Alkane and PAH species don't have high diesel or gasoline emissions; I presume what is meant is that alkane IVOC emission were highest from diesel and PAH emissions highest from gasoline.

We have followed the suggestion of the reviewer and rephrased this sentence to avoid confusion.

**(6)** L209: Provide a rationalisation for the value of 2% being the amount of emissions removed from the ALK5 lumped species.

This is a valid point. n-Dodecane was originally included in ALK5 and represented 2% of this lumped species. In the new model, n-dodecane is included in ALK6 and therefore its emissions are part of this lumped species now. To avoid the double counting of these emissions we have

reduced by 2% the ALK5 emissions. We do explain this small change in the emissions in the revised paper.

**(7)** Caption of Fig. 1: Include in the caption text that average concentration is for May 2008. Also, technically the data are being displayed as mixing ratios, not concentrations.
We have changed the caption of Figure 1 following the suggestions of the reviewer.

**(8)** L252 and caption of Figure 3: Clarify what is meant by "ground-level IVOC concentration" here – does this mean the 7 new lumped species added together?
We have revised the figure caption to indicate that this is the sum of the concentrations of the 7 new lumped species for PMCAMx-iv and the sum of the concentrations of the VBS surrogate species in the IVOC volatility bins for PMCAMx. The same changes have been made to the caption of Figure S1 that shows the corresponding average diurnal profiles.

**(9)** Captions of Figs. 4 & 5: The text could be phrased better to indicate that the data being presented are the on-road transport SOA-iv within the $PM_{2.5}$ size fraction. From the present phrasing, the reader is first given the impression that it is $PM_{2.5}$ data that are being plotted.
We have made the recommended changes to avoid confusion.

**Editorial**
**(10)** L36: use comma rather than semi-colon.
We have made the change.

**(11)** L60: hyphenate "in-field".
We have corrected the typo.

**References**
Fang, H., Huang, X., Zhang, Y., Pei, C., Huang, Z., Wang, Y., Chen, Y., Yan, J., Zeng, J., Xiao, S., Luo, S., Li, S., Wang, J., Zhu, M., Fu, X., Wu, Z., Zhang, R., Song, W., Zhang, G., Hu, W., Tang, M., Ding, X., Bi, X., and Wang, X.: Measurement report: Emissions of intermediate-volatility organic compounds from vehicles under real-world driving conditions in an urban tunnel, Atmos. Chem. Phys., 21, 10005-10013, https://doi.org/10.5194/acp-21-10005-2021, 2021.

Zhao, Y., Hennigan, C. J., May, A. A., Tkacik, D. S., de Gouw, J. A., Gilman, J. B., Kuster, W. C., Borbon, A., and Robinson, A. L.: Intermediate-volatility organic compounds: a large source of secondary organic aerosol, Environ. Sci. Technol., 48, 13743-13750, https://doi.org/10.1021/es5035188, 2014.

Zhao, Y., Nguyen, N. T., Presto, A. A., Hennigan, C. J., May, A. A., and Robinson, A. L.: 715 Intermediate volatility organic compound emissions from on-road diesel vehicles: chemical composition, emission factors, and estimated secondary organic aerosol production, Environ. Sci. Technol., 49, 11516-11526, https://doi.org/10.1021/acs.est.5b02841, 2015.

Zhao, Y., Nguyen, N. T., Presto, A. A., Hennigan, C. J., May, A. A., and Robinson, A. L.: Intermediate volatility organic compound emissions from on-road gasoline vehicles and

small off-road gasoline engines, Environ. Sci. Technol., 50, 4554-4563, https://doi.org/10.1021/acs.est.5b06247, 2016.